# Functional Tensors for Probabilistic Programming

**Fritz Obermeyer**    **Eli Bingham**    **Martin Jankowiak**    **Du Phan**    **Jonathan P. Chen**
Uber AI

## Abstract

It is a significant challenge to design probabilistic programming systems that can accommodate a wide variety of inference strategies within a unified framework. Noting that the versatility of modern automatic differentiation frameworks is based in large part on the unifying concept of tensors, we describe a software abstraction—functional tensors—that captures many of the benefits of tensors, while also being able to describe continuous probability distributions. We demonstrate the versatility of functional tensors by integrating them into the modeling frontend and inference backend of the Pyro probabilistic programming language. As an example application, we perform approximate inference on a switching linear dynamical system.[1]

## 1 Introduction

Probabilistic programming systems allow specification of probabilistic models in high-level programming languages and provide some level of automation for probabilistic inference [1]. It remains a significant challenge to design systems that can accommodate a wide variety of inference strategies—from MCMC to variational inference and beyond—within a unified framework. This work is motivated by the general goal of enabling mixed inferences strategies for probabilistic programs. As a concrete example consider an inference algorithm that exploits modern black-box variational inference as well as classic algorithms that leverage conjugacy (e.g. the Kalman filter). Enabling the former requires support for Monte Carlo sampling and automatic differentiation, while the latter calls for a symbolic computation of sums (for discrete factors) and integrals (for Gaussian factors). To address these requirements, we propose *functional tensors*, a software abstraction that generalizes the algebraic properties of tensors to a wide class of continuous and discrete probability distributions, thus enabling a wide variety of mixed inference strategies in probabilistic programming systems.[2]

## 2 Functional Tensors

Tensors, or more properly "multidimensional arrays", are a popular and versatile software abstraction for performing parallelizable operations on large homogeneous contiguous blocks of memory. Each tensor is backed by a single block of memory, and that memory can be addressed by a tuple of bounded integers, where each integer indexes into a *dimension* of the tensor. Tensor libraries provide operations that act on tensors, including pointwise operations like addition and multiplication, reduction operations such as product and sum, and combined operations such as matrix multiplication and convolution. A useful property of tensor operations is support for *broadcasting*, whereby an operation defined on smaller tensors or scalars can be uniquely extended to an operation on tensors with extra dimensions on the left, so long as all dimensions are of compatible shape.[3]

---

[1] The funsor library and experiments are available at `https://github.com/pyro-ppl/funsor`

[2] See Appendix B for a discussion of related work.

[3] A set of tensors is of *compatible shape* iff, aligning their shapes on the right, for each dimension counting from the right, the set of sizes along each dimension contains no more than a single integer greater than 1.

The observation motivating functional tensors is that tensor dimensions can be viewed as *free variables*, batched tensors can be viewed as *open terms*, and operator support for broadcasting can be viewed as extending ground operators to *open terms*, i.e. terms with free variables [2]. This interpretation of tensors is exploited by the Pyro probabilistic programming language [3] and its implementation of tensor variable elimination for exact inference in discrete latent variable models [4].

Functional tensors (hereafter "**funsors**") generalize tensors by allowing free variables to be of type not only bounded integer, but also other types that appear in probabilistic models, such as real number, real-valued vector, or real-valued matrix. While in general there is no finite representation of functions of real variables, we provide an interface for restricted classes of functions (or properly distributions), including lazy algebraic expressions, non-normalized Gaussian functions, and Dirac delta distributions.

The remainder of this section is organized as follows: in Sec. 2.1-2.2 we overview funsor syntax and operational semantics; in Sec. 2.3 we illustrate funsor usage in probabilistic programming 2.3; and in Sec. 2.4 we describe the atomic distribution funsors Tensor, Gaussian, and Delta.

## 2.1 Funsor syntax

Funsors are terms in a first order language of arrays and array indices; we exclude higher order functions.

**Definition 1.** A *type* is defined by the grammar

$$
\begin{aligned}
\tau \in \text{Type} ::= \;& \mathbb{Z}_n && \text{"bounded integer"} \\
& \big|\; \mathbb{Z}_{n_1} \times \cdots \times \mathbb{Z}_{n_k} \to \mathbb{R} && \text{"real-valued array"}
\end{aligned}
$$

for any $n, n_1, \ldots, n_k, k \in \mathbb{N}$. A *type context* is a list $\Gamma = (v_1{:}\tau_1, \ldots, v_k{:}\tau_k)$ of name:type pairs for names $v \in \mathbb{S}$ in a countable set of symbols $\mathbb{S}$ and types $\tau$.

**Definition 2.** A *funsor* is defined by the grammar

$$
\begin{aligned}
e \in \text{Funsor} ::= \;& \text{Tensor}(\Gamma, w) && \text{"discrete factor"} & \big|\; & \widehat{f}(e_1, \ldots, e_n) && \text{"apply function"} \\
& \big|\; \text{Gaussian}(\Gamma, i, P) && \text{"Gaussian factor"} & \big|\; & e_1[v = e_2] && \text{"substitute"} \\
& \big|\; \text{Delta}(v, e) && \text{"point mass"} & \big|\; & \textstyle\sum_v e && \text{"marginalize"} \\
& \big|\; \text{Variable}(v, \tau) && \text{"delayed value"} & \big|\; & \textstyle\prod_v e && \text{"product"}
\end{aligned}
$$

where $\Gamma$ is a type context, $w, i, P$ are numerical arrays, $v \in \mathbb{S}$ is a variable name, $\tau \in T$ is a type, and $f$ is any function defined on numerical objects, e.g. binary multiplication $e_1 \times e_2$ and nullary constants 0 and 1 for each type.

**Definition 3.** The set of free variables of a funsor $e$ is denoted $\text{fv}(e)$. A funsor is *open* if it has free variables and *closed* otherwise. Each basic numerical object $x \in \tau$ defines a *ground* funsor $\widehat{x} = \text{Tensor}((), x)$.

## 2.2 Operational semantics

Funsor computations are executed by seminumerical term rewriting. We specify a set of rewriting rules and rely on a dispatch mechanism to match and execute rules until termination. Each rule contains a pattern and behavior to perform on match. The behavior includes both symbolic term rewriting and low-level numerical computation, similar to operations in tensor libraries for automatic differentiation (AD). In contrast to tensor operations in AD libraries, funsor expressions can be non-analytic, in which case they can only be evaluated approximately.

To support approximation of non-analytic funsor expressions and optimization of large funsor expressions, we rely on nonstandard interpretation [6, 7]. Each interpretation is a set of rewrite rules, and interpretations can be selected and interleaved at runtime. For example an EXACT interpretation eagerly evaluates tractable funsors but leaves non-analytic integrals lazy; a fully LAZY interpretation records an expression for optimization and static analysis; and MONTE CARLO and MOMENT MATCHING interpretations add extra rules for approximate evaluation of integrals.

## 2.3 Application to probabilistic programming

Funsors fill two roles in probabilistic programming: as compute graphs for lazy tensor computations in user-facing model code, and as seminumerical representations of joint distributions in automatic inference strategies. In the following we consider two probabilistic inference tasks that leverage delayed sampling [8] in detail.

Figure 1 illustrates a funsor computation for MAP inference in a simple generative model. Inference steps on the right are triggered by execution of each line of model code on the left. On line 1 the joint distribution is initialized to the trivial normalized distribution. On line 2 a delayed sample statement triggers creation of a new free variable $z$ in the model code and accumulation of an unevaluated factor distribution $P_z[v = z]$ in inference code. (We assume by convention distributions like $P_z$, $P_x$, $Q$ name their variate $v$ and parameter, if any, $\theta$.) On line 3 a nonlinear function is lazily applied to $z$ creating a lazy funsor expression $y$. On line 4 a distribution is conditioned on ground data $x$, triggering accumulation of a factor $P_x[\theta = y, v = x]$ with free variable $z$ (because $x$ is ground and $y$ has free variable $z$). Model termination on line 5 triggers marginalization of the $x$ variable, which can be performed either exactly by pattern matching or approximately by Monte Carlo sampling. The resulting objective is differentiable with respect to any parameters. Optimization is achieved by stochastic gradient ascent, repeatedly executing model code and accumulating factors.

Figure 2 illustrates a typical funsor computation in variational inference, where a data-dependent variational distribution $Q$ is fit to data. Lines 1–6 execute delayed sample statements and accumulate distributions $p$ and $q$ with a single free variable $z$. Line 7 computes the ELBO, which can be performed either exactly by pattern matching or approximately by Monte Carlo sampling $z$ from $Q$.

```
1 fun GenerativeModel(x)      p ← 1
2     z ← sample(P_z)          p ← p × P_z[v = z]
3     y ← exp(z)
4     observe(P_x[θ = y], x)   p ← p × P_x[θ = y, v = x]
5 end                          maximize: Σ_z p
```

**Figure 1:** User-facing probabilistic program (left) and automatic inference (right) for (delayed) MAP inference.

```
1 fun GenerativeModel(x)      p ← 1
2     z ← sample(P_z)          p ← p × P_z[v = z]
3     observe(P_x[θ = z], x)   p ← p × P_x[v = x, θ = z]
4 end
5 fun InferenceModel(x)       q ← 1
6     z ← sample(Q[θ = x])     q ← q × Q[v = z, θ = x]
7 end                          maximize: Σ_z q log p/q
```

**Figure 2:** User-facing probabilistic program (left) and automatic inference (right) for variational inference with delayed sampling. The quantity maximized is the ELBO.

Both of the above delayed sampling computations (MAP and ELBO) proceed by first building a large sum-product expression[4] and then evaluating this expression through a combination of pattern matching and approximation. An alternative to delayed sampling is eager sampling, where `sample` statements in the model trigger Monte Carlo sampling, no free variables are created, and marginalization $\sum_z$ is not needed. Funsors allow eager and delayed sampling to be combined freely.

## 2.4 Numerics of distribution funsors

Distribution funsors are the basic latent factors in sum-product expressions constructed during probabilistic inference. While we implement a large number of distribution funsors to serve as likelihoods in `observe` statements, we focus attention on three atomic distributions that are closed under sums and products, and thus especially attractive as distributions for latent variables. These three special funsors are: i) Tensor funsors to represent non-normalized discrete joint probability mass functions; ii) Gaussian funsors to represent non-normalized joint multivariate normal distributions among a set of real-tensor valued variables, possibly dependent on other discrete variables; and iii) Delta funsors to represent degenerate distributions and Monte Carlo samples.

**Tensor funsors** represent a non-normalized mass function as a single tensor (multidimensional array) of weights. Thus standard variable elimination can be seen as mere tensor contraction. Memory cost and computation cost are both exponential in the number of free variables. The crucial rewrite rule for

---

[4]For clarity this paper uses the $(+, \times)$ semiring, but our implementation performs inference computations in log-space using the $(\mathrm{logaddexp}, +)$ semiring.

Tensor funsors allows operations $f(e_1, \ldots, e_n)$ on Tensor funsors $e_1, \ldots, e_n$ to be eagerly evaluated even in the presence of free variables; this is especially useful when e.g. $f$ is a neural network whose inputs depend on lazily sampled discrete random variables:

$$\widehat{f}(\text{Tensor}, \ldots, \text{Tensor}) \Rightarrow \text{Tensor} \qquad \text{``batched apply''}$$

**Gaussian funsors** represent a log density function among multiple real-tensor-valued free variables using the information form of the Kalman filter [9, 10], i.e. as pair $(v, P)$, where $v = P\mu$ is the information vector, $\mu$ is the mean, and $P = \Sigma^{-1}$ is the precision matrix, the inverse of the covariance matrix $\Sigma$. The information form is useful in information fusion problems because it allows representation of rank-deficient joint distributions, such as a conditional distribution treated as a single Gaussian factor; in practice a joint distribution often becomes full rank only after fusing multiple individually rank-deficient Gaussian factors. Memory cost is quadratic and computation cost is cubic in the total number of elements in all free real-tensor-valued variables; both costs are exponential in the number of bounded integer free variables.

**Delta funsors** represent a normalized point distribution as a pair $(v, x)$, where $v$ is a symbol and $x$ is a Tensor funsor, possibly with free discrete variables corresponding to batch dimensions. The crucial rewrite rule for Delta funsors triggers substitution:

$$\text{Delta}(v, e_1) \times e_2 \Rightarrow \text{Delta}(v, e_1) \times e_2[v = e_1] \qquad \text{if } v \in \text{fv}(e_2)$$

Tensors, Gaussians, and Deltas are algebraically closed in combination, i.e. any sum-product of Tensor, Gaussian, and Delta factors can be rewritten[5] to a product of zero or more deltas, an optional Tensor, and an optional Gaussian. Our rewrite system captures this fact as a "joint normal form" funsor representing a lazy finitary product, together with rules for commutativity, associativity, distributivity, and substitution.

## 3 Experiment: Switching Linear Dynamical System

To demonstrate the versatility of funsors as a substrate for probabilistic programming, we perform approximate inference on a switching linear dynamical system [11], leveraging a moment-matching approximation to make inference tractable. See Appendix A for details.

## 4 Conclusion

We introduced *funsors*, a software abstraction that generalizes tensors to provide finite representations for a restricted class of discrete and continuous distributions, including lazy algebraic expressions, non-normalized Gaussian distributions, and Dirac delta distributions. We demonstrated how funsors can be integrated into a probabilistic programming system, thereby enabling a wide variety of inference strategies. In future work we will describe how funsors can be used to represent generalized variable elimination [4] and parallel-scan filtering algorithms [14] that enable parallel exact inference for a large family of structured probabilistic models.

### Acknowledgements

We thank Ruy Ley-Wild, Mahmoud Abokhamis, Hung Ngo, Noah Goodman and Alexander M. Rush for helpful discussions.

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

|       | L = 1 |       | L = 3 |       | L = 5 |       |
|-------|-------|-------|-------|-------|-------|-------|
| Model | MSE   | LL    | MSE   | LL    | MSE   | LL    |
| SLDS-I   | 0.574 | -10.13 | 0.574 | -10.13 | 0.574 | -10.13 |
| SLDS-II  | 0.527 | -9.55  | 0.497 | -9.64  | 0.498 | -9.64  |
| SLDS-III | 0.512 | **-9.33** | 0.511 | -9.41  | **0.482** | -9.46  |

**Table 1:** Test log likelihoods and mean squared errors for SLDS variants with various moment-matching window lengths $L$. See Sec. A for details.

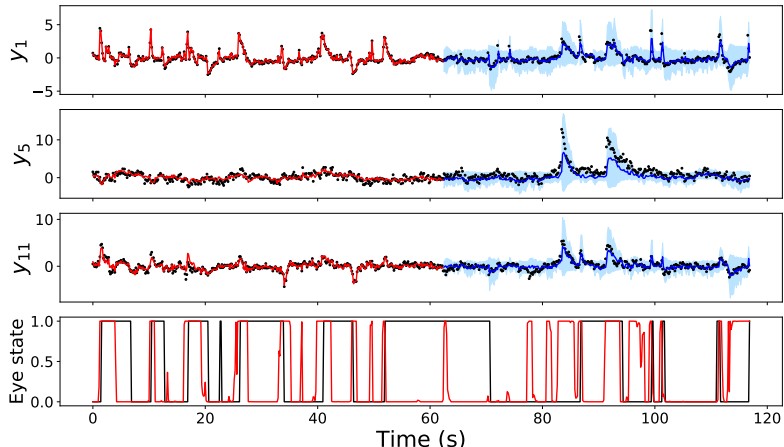

**Figure 3:** Smoothing estimates (red) and 1-step-ahead ahead predictions (blue; with 90% confidence intervals) for three randomly selected output dimensions ($y_1, y_5, y_{11}$) for the SLDS experiment in A. The bottom figure depicts the observed eye state (black) as well as the smoothing estimate of the inferred switching label $s_t$ (red).

## A  Switching Linear Dynamical System

We use a switching linear dynamical system (SLDS) [11] to model an EEG time series dataset $\{y_t\}_{t=1}^T$ from the UCI database [12]. The generative model is as follows. At each time step $t$ there is both a discrete switching label $s_t \in [1, ..., K]$ and a continuous latent state $x_t$; both follow Markovian dynamics, see Fig. 4. We consider three model variants: I) the transition probabilities $p(x_t|x_{t-1}, s_t)$ depend on the switching state; II) the emission probabilities $p(y_t|x_t, s_t)$ depend on the switching state; and III) both the transition and emission probabilities depend on the switching state.

Exact inference for this class of models is $\mathcal{O}(K^T)$. To make inference tractable, we use a moment-matching approximation with window length $L$, reducing the complexity to $\mathcal{O}(K^{L+1})$. Representing this approximate inference algorithm follows immediately by employing a `moment_matching` interpretation for funsor reductions. For parameter learning we use gradient ascent on the (approximate) log marginal likelihood $\log p(y_{1:T})$. See Table 1 for the results we obtain for all three model variants with $K = 2$ switching states and window lengths $L \in \{1, 3, 5\}$. We obtain the best results with the richest model (SLDS-III), with the most expensive moment-matching approximation ($L = 5$) yielding the lowest mean squared error.

In Fig. 3 we depict smoothing estimates for the training data and one-step-ahead predictions for the held-out data using the best performing model, validating the efficacy of the moment-matching approximation. The EEG data also include an observed eye state (0: open, 1: closed) at each time step. We note that the transitions between switching states in the learned model correlate reasonably well with eye state transitions, despite the fact that the model did not have access to observed eye states during training.

The joint probability $p(y_{1:T}, s_{1:T}, x_{1:T})$ of model variant SLDS-I is given by

$$\prod_{t=1}^{T} p(s_t|s_{t-1})\mathcal{N}(x_t|A^{s_t}x_{t-1}, \sigma^{s_t}_{\text{trans}})\mathcal{N}(y_t|Bx_t, \sigma_{\text{obs}})$$

where $A^{s_t}$ is a state-dependent transition matrix, $\sigma^{s_t}_{\text{trans}}$ is a state-dependent diagonal transition noise matrix, B is a state-independent observation matrix, and $\sigma_{\text{obs}}$ is a state-independent diagonal

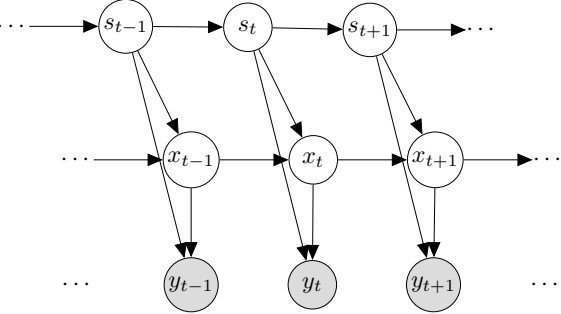

**Figure 4:** Graphical structure for the SLDS-III model in Sec. A. The $\{s_t\}$ form a chain of discrete switching states and the $\{x_t\}$ are continuous. Here the transition probabilities $p(x_t|x_{t-1}, s_t)$ and emission probabilities $p(y_t|x_t, s_t)$ both depend on the switching label $s_t$.

observation noise matrix. Similarly, the joint probability of variant SLDS-II is given by:

$$\prod_{t=1}^{T} p(s_t|s_{t-1})\mathcal{N}(x_t|Ax_{t-1}, \sigma_{\text{trans}})\mathcal{N}(y_t|B^{s_t}x_t, \sigma_{\text{obs}}^{s_t})$$

where now $A$ and $\sigma_{\text{trans}}$ are state-independent and $B^{s_t}$ and $\sigma_{\text{obs}}^{s_t}$ are state-dependent. Finally, the joint probability of variant SLDS-III is given by

$$\prod_{t=1}^{T} p(s_t|s_{t-1})\mathcal{N}(x_t|A^{s_t}x_{t-1}, \sigma_{\text{trans}}^{s_t})\mathcal{N}(y_t|B^{s_t}x_t, \sigma_{\text{obs}}^{s_t})$$

where now both the transition and emission probabilities are state-dependent. In all our experiments we use $K = 2$ switching states and set the dimension of the continous state to $\dim(x_t) = 5$.

To compute the log marginal likelihood used in training we use a moment-matching approximation with a window length of $L$, see Ex. 5. During prediction and smoothing we use $L = 1$.

The raw dataset has $T = 14980$ timesteps, which we subsample by a factor of 20, yielding a dataset with $T = 749$. We use the first 400 timesteps for training. Of the remaining 349 timesteps, we use random subsets of size 149 and 200 for validation and testing, respectively. In particular we use the validation set to choose learning hyperparameters and determine early stopping for gradient ascent. The 14-dimensional outputs $\{y_t\}$ are normalized to have zero mean and unit variance.

We use the Adam optimizer for training [13]. We train for up to 200 gradient steps and decay the learning rate exponentially. We use the validation set to do a hyperparameter search over the exponential decay factor $\gamma$ and the momentum parameter $\beta_1$. For each hyperparameter setting we do 7 independent runs with different random number seeds for parameter initialization. We then report results on the test set.

```
# returns the marginal log probability of the observed data. we use an interpretation decorator to
# signal to funsor that all reduce operations should be done using a moment—matching approximation.
#
# inputs:
# observations (torch.Tensor of shape (T, obs_dim))
# trans_probs, x_init_dist, x_trans_dist, y_dist (funsors)
@funsor.interpreter.interpretation(funsor.terms.moment_matching)
def marginal_log_prob(observations, trans_probs, x_init_dist, x_trans_dist, y_dist,
                      L=2, num_components=2, hidden_dim=5):
    log_prob = funsor.Number(0.)
    s_vars = {—1: funsor.Tensor(torch.tensor(0), dtype=num_components)}
    x_vars = {}

    for t, y in enumerate(observations):
        s_vars[t] = funsor.Variable(f's_{t}', funsor.bint(num_components))
        x_vars[t] = funsor.Variable(f'x_{t}', funsor.reals(hidden_dim))

        # incorporate discrete switching probability p(s_t | s_{t—1})
        log_prob += dist.Categorical(trans_probs(s=s_vars[t — 1]), value=s_vars[t])

        # incorporate continuous transition probability p(x_t | x_{t—1}, s_t)
        if t == 0:
            log_prob += x_init_dist(value=x_vars[t])
        else:
            log_prob += x_trans_dist(s=s_vars[t], x=x_vars[t — 1], y=x_vars[t])

        # do a moment—matching reduction of latent variables from L time steps in the past
        # [i.e. we retain a running (L+1)—length window of latent variables throughout the for loop]
        if t > L — 1:
            log_prob = log_prob.reduce(ops.logaddexp, {s_vars[t — L].name, x_vars[t — L].name})

        # incorporate observation probability p(y_t | x_t, s_t)
        log_prob += y_dist(s=s_vars[t], x=x_vars[t], y=y)

    T = data.shape[0]
    for t in range(L):
        log_prob = log_prob.reduce(ops.logaddexp, {s_vars[T — L + t].name, x_vars[T — L + t].name})

    return log_prob
```

**Figure 5:** `funsor` code for the computation of the log marginal probability $\log p(y_{1:T})$ for the SLDS model in Sec. A.

## B Related Work

PSI Solver [17] implements exact inference algorithms for probabilistic programs using symbolic algebra. Hakaru [18, 19] implements a probablistic program inference optimizer that compiles to Maple expressions for symbolic manipulation, then performs MCMC inference. Our work can be seen as a mixed symbolic-numerical approach that provides limited symbolic pattern manipulation and relies on a high-level tensor library (PyTorch [20] or JAX [21]) for automatic differentiation and parallelization. Indeed we see functional tensors as a compromise between fully symbolic and fully numerical integration in the same way that automatic differentiation is a compromise between symbolic differentiation and numerical differentiation [22].

Dillon et al. [23] describe a low-level software abstraction for implementing probability distributions, in particular taking care to implement batching and broadcasting. Our work can be seen as generalization of such distributions in three directions: from broadcastable dimensions to free variables, from normalized to non-normalized, and from single distributions to joint distributions (still with $O(1)$ underlying tensors). Hoffman et al. [24] design a system for automatic conjugacy detection in stochastic computation graphs; our system matches coarser patterns, e.g. Gaussians rather than polynomials.

A number of inference algorithms naturally generalize to funsors. Obermeyer et al. [4] generalize variable elimination to factor graphs with plates, proving that plated factor graphs can be easily separated into those in which the complexity of discrete variable elimination grows either exponentially or linearly in plate size. Särkkä and García-Fernández [14] adapt parallel scan algorithms to Bayesian filtering settings, demonstrating exponential parallel speedup in inference in common probabilistic graphical modeling methods such as discrete state hidden Markov models and Kalman filters. Bilmes [15] leverages repeated (typically dynamic) structure in graphical models to quickly compute a sequential variable elimination schedule. [16] develop bounded memory inference algorithms for sequential probabilistic models.

