# OpenReview forum: "Functional Tensors for Probabilistic Programming"
_NeurIPS.cc/2019/Workshop/Program_Transformations — Program Transformations @NeurIPS2019 Oral_

### Official Review · AnonReviewer2 · 2019-09-29
**Functional tensors (funsors) for probabilistic programming**

**Confidence:** 4
**Rating:** 8

**Review:**

This paper introduces the idea of “functional tensors” (funsors). A funsor is a generalization of a tensor (multi-dimensional array) data type by viewing tensor dimensions as free variables in lambda calculus. In the introduced framework, funsors can represent tensors (discrete factors), Gaussian and Dirac delta factors, variables, function applications, substitutions, marginalizations, and products. The language of funsors is first order in the sense that higher-order functions are excluded. Funsor expressions are executed by term rewriting that can use a number of nonstandard interpretation schemes, namely, exact, lazy, Monte Carlo, and moment matching. The work is motivated by developing probabilistic programming systems based on funsors.

The paper is well-written and structured, but difficult to understand and follow for any reader outside the probabilistic programming and programming languages communities. I fear that this makes the work inaccessible to the wider machine learning community, and the paper would definitely benefit from the addition of some paragraphs to explain why such an approach is relevant or useful in practice.

I learned quite a lot from this paper, and I also like the way nonstandard interpretation examples are presented as side-by-side panels of user-facing programs and corresponding side effects for inference in the examples in Figures 1 and 2. The paper provides a link to code on GitHub repository https://github.com/pyro-ppl/funsor which is definitely helpful in understanding the actual implementation details of the presented work.

I am definitely interested in learning more about this work. I believe it is a very good fit for this workshop.

---

### Official Review · AnonReviewer1 · 2019-09-30
**Unqualified to review but looks good to me**

**Confidence:** 2
**Rating:** 7

**Review:**

This abstract is definitely outside of my area of expertise. That said, the idea of having the types of tensors as free variables looks powerful. Given the interdisciplinary nature of the workshop, this abstract could use a lot of work being made more accessible for people without a PL/PPL background though.

---

### Decision · Program_Chairs · 2019-10-01

**Decision:**

Accept (Oral)

**Comment:**

This is an interesting new idea that we would like to hear more about in an oral presentation. However, the authors did note that this abstract was hard to read for people without a background in probabilistic programming. Considering the interdisciplinary nature of the workshop, we suggest the authors focus on making their work accessible and intuitive for a wider audience.